# Changes in Expression of Complement Components in the Ovine Spleen during Early Pregnancy

**DOI:** 10.3390/ani11113183

**Published:** 2021-11-08

**Authors:** Ling Yang, Luyu Wang, Jiaxuan Wu, Haichao Wang, Gengxin Yang, Leying Zhang

**Affiliations:** School of Life Sciences and Food Engineering, Hebei University of Engineering, Handan 056038, China; zgwly2016@126.com (L.W.); wujiaxuan519@163.com (J.W.); qiusuoba@163.com (H.W.); ygengxin@163.com (G.Y.); zhangly056000@126.com (L.Z.)

**Keywords:** pregnancy, complement pathway, immune regulation, spleen, sheep

## Abstract

**Simple Summary:**

Complement regulation is related to fetal growth during early pregnancy in humans. The results show that C1q increased during early pregnancy, and C1r, C1s, C2, C3 and C5b enhanced at day 25 of gestation, and C4a and C9 increased on days 13 and 16 of pregnancy in the ovine maternal spleen. In summary, a complement pathway participates in maternal immune regulation during early gestation.

**Abstract:**

During early gestation in humans, complement regulation is essential for normal fetal growth. It is supposed that a complement pathway participates in maternal splenic immune regulation at the early stage of gestation in ewes. The aim of this study was to analyze the effects of early pregnancy on the expression of complement components in the maternal spleen of ewes. In this study, ovine spleens were sampled on day 16 of nonpregnancy, and days 13, 16 and 25 of gestation. RT-qPCR, Western blot and immunohistochemical analysis were used to detect the changes in expression of complement components in the ovine maternal spleens. Our results reveal that C1q was upregulated during early gestation, C1r, C1s, C2, C3 and C5b increased at day 25 of gestation and C4a and C9 peaked at days 13 and 16 of gestation. In addition, C3 protein was located in the capsule, trabeculae and splenic cords. In conclusion, our results show for the first time that there was modification in the expression of complement components in the ovine spleen at the early stage of gestation, and complement pathways may participate in modulating splenic immune responses at the early stage of gestation.

## 1. Introduction

A complement pathway is the replenishment of immunological processes, and is implicated in immune surveillance, cell homeostasis and tissue development [1]. The complement system participates in the complex tolerance and clearance processes, and complement system dysregulation results in insufficient clearance of the fragmentation of the placental tissue and preeclampsia in humans [2]. The activity of the complement system is enhanced systemically during pregnancy, but the activity is inhibited at the placenta for pregnancy maintenance [3]. Complement regulation is necessary for a successful pregnancy, and some complement components are favorable in the normal fetal growth at the stage of pre-implantation to placentation [4]. However, during early pregnancy, complement regulation in the spleen is unclear in ewes.

There is selective immunosuppression in the uterus and peripheral tissues induced by conceptus signals, which are essential for the fetus to evade maternal immune detection and elimination in domestic farm animals [5]. The spleen has a wide range of immunologic functions and plays key roles in the initiation of adaptive immunity [6]. Interleukin-33 (IL-33) participates in innate and adaptive immunity, and the IL-33 receptor is upregulated in splenic B cells during normal gestation in mice [7]. Our previous studies demonstrate that interferon-stimulated genes (ISGs), progesterone (P4) receptor, P4-induced blocking factor, tumor necrosis factor β, interleukin-2 (IL-2), IL-5, IL-6, IL-10, cyclooxygenase 2, aldo-keto reductase family 1, member B1, melatonin receptor 1 (MT1), gonadotropin-releasing hormone (GnRH) and its receptor are upregulated. However, MT2 is decreased in the ovine maternal spleen at the early stage of gestation [8,9,10,11,12,13]. It is presumed that a complement pathway is involved in the immune tolerance regulation of the maternal spleen at the early stage of gestation in sheep. The aim of the present research was to analyze the expression of complement components in the maternal spleen at the early stage of gestation in ewes, which may be useful for understanding the establishment of the immune tolerance mechanism of the maternal spleen.

## 2. Materials and Methods

### 2.1. Animals and Experimental Design

The experimental design and tissue collection have been described in detail before [9,10,11]. Briefly, 35 ewes with similar age, BCS and parity received the same diet, and controlled internal drug-releasing devices (InterAg, Hamilton, New Zealand) were used for estrus synchrony. After estrus detection (day 0), six ewes were randomly selected for nonpregnant ewes and slaughtered at day 16 after estrus (day 16 of the estrous cycle, DN16) for spleen collection. The other 29 animals were exposed to fertile rams, which ensured there were at least 18 pregnant ewes. The pregnant animals were randomly killed at days 13, 16 and 25 after mating for spleen collection (*n* = 6 for each group). Days 13, 16, and 25 of pregnancy (DP13, DP16, and DP25), and day 16 of the estrous cycle (DN16), were chosen because the maternal spleen was under the different effects of interferon-tau and/or P4, or not on these days [12]. There was a morphologically normal conceptus for pregnant ewes at slaughter. Cross sections (0.5 cm thick) of the spleens were collected for subsequent mRNA and proteins analysis, and immersed in 4% paraformaldehyde in PBS for subsequent immunohistochemistry.

### 2.2. RT-qPCR Assay

The total RNA isolation and cDNA synthesis have been described in detail before [12]. The specified primers were designed for the ovine (Ovis aries) complement component genes (Appendix A) and synthesized by Shanghai Sangon Biotech Co., Ltd. (Shanghai, China). The primer matrix experiments were used for determining the optimal primer concentrations. The amplification efficiency of each primer sequence was in an acceptable range. A SuperReal PreMix Plus kit (Tiangen Biotech, Beijing, China) was used for quantitative PCR in a CFX96 real-time PCR system (Bio-Rad, Hercules, CA, USA), and *GAPDH* was applied as a housekeeping gene. Relative transcript abundances of complement component genes were analyzed utilizing the 2^−ΔΔCt^ analysis method [14]. The mean CT value of DN16 was applied to normalize the mRNA expression level of DP13, DP16 and DP25.

### 2.3. Western Blot

The proteins were isolated from ovine splenic samples as described previously [13]. Samples were isolated by SDS-polyacrylamide gel electrophoresis and transferred electrophoretically on PVDF membranes (Millipore, Bedford, MA, USA). The membranes were blocked with skim milk, and incubated with the antibodies (Appendix A) at a dilution of 1:1000. Then, the membranes were incubated with an HRP-conjugated anti-mouse secondary antibody (Biosharp, Hefei, China BL001A, 1:2000). An HRP chemiluminescence kit was applied for detecting blots. The blot intensity was analyzed by Quantity One V452 (Bio-Rad Laboratories)., Hercules, CA, USA). The relative intensity of the blots was calculated and normalized with a reference protein (GAPDH) using an anti-GAPDH antibody (Appendix A, 1:1000).

### 2.4. Immunohistochemical Analysis

Splenic tissue was dehydrated and embedded in paraffin wax. The paraffin-embedded sample was sectioned in 5 μm-thick sections, mounted onto slides, deparaffinized and rehydrated. Sections underwent antigen recovery in boiling 0.01 M citric buffer and were treated with 3% H_2_O_2_ for blocking endogenous peroxidase activity. After blocking nonspecific binding sites using 5% normal goat serum in PBS, the sections were incubated with the anti-C3 antibody (Appendix A, 1:200). The negative control sections were incubated with an antiserum-specific isotype instead of the anti-C3 antibody. The specific binding site was detected utilizing a DAB kit (Tiangen Biotech). In the end, a light microscope (Nikon Eclipse E800, Tokyo, Japan) with a digital camera DP12 was applied for image capture, and the photo was analyzed independently by four investigators. The staining intensities of the splenic samples were analyzed through the images in a blind manner and assigned an immunoreactive intensity (from 1 to 3) [11].

### 2.5. Statistical Analysis

Least-squares ANOVA applying Mixed and General Linear Model procedures of the Statistical Analysis System (SAS Institute, Cary, NC, USA) was applied to analyze the relative abundance levels of mRNA and protein. The PROC UNIVARIATE procedure (SAS Institute Inc.) was used for testing the data normality, and the non-parametric test was applied in cases not achieving normal distribution. Data were expressed as least squares means. Data obtained from the different spleens of animals were analyzed for the main effects of day and pregnant status and the interaction between the main effects. The Duncan method was applied to analyze multiple comparisons. A *p*-value < 0.05 was considered significantly different.

## 3. Results

### 3.1. Expression of Complement Component mRNA in the Spleen

It is revealed in Figure 1 that the *C1q* mRNA increased significantly during early pregnancy, and reached the highest level at DP25 (Figure 1; *p* = 0.0023 between DN16 and DP25; *p* = 0.0087 between DP13 and DP25; *p* = 0.0063 between DP16 and DP25). The relative values of *C1r*, *C1s*, *C2*, *C3* and *C5b* mRNA upregulated at DP25 (*p* = 0.0034 between DN16 and DP25, *p* = 0.0037 between DP13 and DP25 and *p* = 0.0035 between DP16 and DP25 for *C1r*; *p* = 0.0082 between DN16 and DP25, *p* = 0.0076 between DP13 and DP25 and *p* = 0.0021 between DP16 and DP25 for *C1s*; *p* = 0.0071 between DN16 and DP25, *p* = 0.0065 between DP13 and DP25 and *p* = 0.0073 between DP16 and DP25 for *C2*; *p* = 0.0075 between DN16 and DP25, *p* = 0.0081 between DP13 and DP25 and *p* = 0.0036 between DP16 and DP25 for *C3*; *p* = 0.0087 between DN16 and DP25, *p* = 0.0079 between DP13 and DP25 and *p* = 0.0032 between DP16 and DP25 for *C5b*), but *C1s*, *C3* and *C5b* mRNAs were downregulated at DP16. In addition, the relative levels of *C4a* and *C9* peaked at DP13 and DP16, but declined at DP25 in the maternal spleen (*p* = 0.0053 between DP13 and DP25, and *p* = 0.0061 between DP16 and DP25 for *C4a*; *p* = 0.0069 between DP13 and DP25 and *p* = 0.0064 between DP16 and DP25 for *C9*).

### 3.2. Protein Expression of Complement Component 9 in the Spleen

Figure 2 and Appendix A show that early gestation stimulated the expression of C1q protein (*p* = 0.0017 between DN16 and DP13; *p* = 0.0092 between DN16 and DP16; *p* = 0.0081 between DN16 and DP25). The values of C1r, C1s, C2, C3 and C5b proteins upregulated at DP25 (*p* = 0.0061 between DN16 and DP25, *p* = 0.0059 between DP13 and DP25 and *p* = 0.0068 between DP16 and DP25 for C1r; *p* = 0.0089 between DN16 and DP25, *p* = 0.0097 between DP13 and DP25 and *p* = 0.0046 between DP16 and DP25 for C1s; *p* = 0.0031 between DN16 and DP25, *p* = 0.0076 between DP13 and DP25 and *p* = 0.0083 between DP16 and DP25 for C2; *p* = 0.0037 between DN16 and DP25, *p* = 0.0043 between DP13 and DP25 and *p* = 0.0014 between DP16 and DP25 for C3; *p* = 0.0049 between DN16 and DP25, *p* = 0.0047 between DP13 and DP25 and *p* = 0.0028 between DP16 and DP25 for C5b). In addition, the C4a and C9 proteins were expressed at DP13 and DP16 (*p* = 0.0034 between DP13 and DP25, and *p* = 0.0029 between DP16 and DP25 for *C4a*; *p* = 0.0051 between DP13 and DP25, and *p* = 0.0048 between DP16 and DP25 for *C9*). However, it was undetected at DN16 and DP25.

### 3.3. Immunohistochemical Location for C3 Protein

C3 protein was limited to the capsule, trabeculae and splenic cords (Figure 3). The staining intensities in the splenic samples for C3 were zero (negative), two (strong), two (weak), one (strong), and three (stronger) for the negative control, the spleens from DN16 and spleens from DP13, DP16 and DP25, respectively (Figure 3).

## 4. Discussion

### 4.1. Early Pregnancy Changed the Expression of Complement Components in the Maternal Spleen

Our data reveal that C1q expression was upregulated during the early stage of gestation. Paternal deficiency of C1q results in fetal growth restriction in a preeclampsia-like gestation in mice [15]. The complement protein C1q is expressed in gestational tissues during pregnancy, which is involved in regulating the mother’s immune responses [16]. Therefore, the increase in C1q during early gestation participates in an adjustment of the maternal immune responses that is essential for pregnancy maintenance in ewes.

There were increases in C1r and C2 in the maternal spleen at DP25 in this study. In cattle, there is an upregulation of *C1r* mRNA in the luminal and glandular epithelial cells during the preattachment period [17]. C1r is undetected in the implantation site of the preeclampsia model [18], suggesting that C1r expression in the implantation site is beneficial for successful pregnancy in mice. C2-deficient patient is susceptible to developing to thrombocytopenia during early gestation [19]. Therefore, the increases in C1r and C2 in the maternal spleen may be necessary for normal early gestation in ewes.

C1s and C3 declined at DP16, but increased at DP25 in this study. Amnion tissue explants can synthesize C1s, which contribute to the local host defense in humans [20]. C1s protein is expressed in chorionic tissue, and secreted extracellularly, which is related to the tolerance of the allograft fetus in humans [21]. In metazoans, C3 interacts with many complement factors and non-complement proteins to play an essential role in immune regulation [22]. C3 protein is downregulated significantly in the serum of pregnant women with neural tube defects compared with the women carrying normal fetuses during pregnancy [23]. Plasma concentration of the C3 protein is increased gradually throughout pregnancy compared with nonpregnant women, suggesting that C3 plays a key role in normal placentation [24]. Therefore, it is possible that upregulation of C1s and C3 at DP25 may be related to placentation.

Our data reveal that C5b was downregulated at DP16, but upregulated at DP25. C5 is cleaved to generate C5b and C5a by the C3-cleaving enzyme, and C5b participates in immune responses through assembling with downstream complement components [25]. C5b-9 is localized in the surface of syncytiotrophoblasts, intervillous fibrin and decidual vessels, which contributes to placental formation [26]. There is a higher level of C5b-9 in the villous trophoblast of placentas from normal pregnant women compared with the patients with preeclampsia [27]. Therefore, the increase in C5 at DP25 may be essential for placentation.

C4a and C9 peaked at days 13 and 16 of pregnancy in this study. Human chorionic gonadotropin stimulates the C4a gene expression in the baboon endometrium, which is implicated in regulating the decidual immune environment at the time of implantation [28]. The frequency of C4 ‘null’ alleles is increased in recurrent spontaneous abortions, suggesting that C4a is necessary for a successful pregnancy in humans [29]. However, there is a high concentration of circulating C4 in preeclamptic patients compared with that in normal nonpregnant women [30]. C9 is the major component of the membrane attack complex that forms pores in the plasma membrane of target cells to participate in the adaptive immune response [31]. After the first trimester of gestation, the serum level of C9 is downregulated in women with normal pregnancy compared with females with preeclampsia [32]. Therefore, it is suggested that the upregulation of C4a and C9 on DP13 and DP16 is related to implantation, and the downregulation of C4a and C9 at DP25 is helpful for pregnancy maintenance.

The spleen participates in the blood defense against blood-borne pathogens, and is also partly implicated in the immune responses [33]. The immunohistochemistry for C3 proteins was mainly located in the capsule, trabeculae and splenic cords, and the staining intensities for C3 protein were changed during early pregnancy. There is a change in the activation state of peripheral CD4 lymphocytes in the maternal spleen during the preimplantation period of gestation in mice [34]. Early gestation changed the expression of the P4 receptor, ISGs, prostaglandin synthases, T helper cytokines, MTs, GnRH and its receptor in the ovine maternal spleen in our previous papers [8,9,10,11,12,13]. C3 plays key roles in three complement pathways and is directly related to immune regulation and immune defense activation [22]. Therefore, it is suggested that a complement pathway is related to modifying maternal splenic immune tolerance through C3-dependent signaling during early gestation in ewes.

### 4.2. Possibility for Clinical Implications

In this study, there was upregulation of C1r, C3 and C5b in the maternal spleen, but the expression of C4 and C9 was downregulated at DP25. C1r expression is decreased in the implantation site of the preeclampsia model [18], and C2 deficiency results in thrombocytopenia during early pregnancy [19]. Neural tube defects in pregnant women lead to downregulation of the C3 protein in the serum compared with normal pregnant women [23]. Normal women have a lower level of C5b-9 in the villous trophoblast of the placentas [27]. However, the concentration of circulating C4 in preeclamptic patients is higher than in normal women [30], and the serum level of C9 is decreased in women with normal pregnancy after the first trimester of gestation [32]. Therefore, it is suggested that the agonists for C1r, C3 and C5b, and the antagonists for C4 and C9 may be used for improving reproductive efficiency after early pregnancy.

## 5. Conclusions

Early pregnancy induces changes in the expression of complement components in the maternal spleen, and these changes can be used as signs of early pregnancy in ewes. Furthermore, the C3 protein was limited to the capsule, trabeculae and splenic cords. In summary, complement pathways may be related to accommodating maternal spleen immune responses at the early stage of gestation, and the detection of C1q, C4a and C9 may be used for early pregnancy diagnosis in ewes.

## Figures and Tables

**Figure 1 animals-11-03183-f001:**
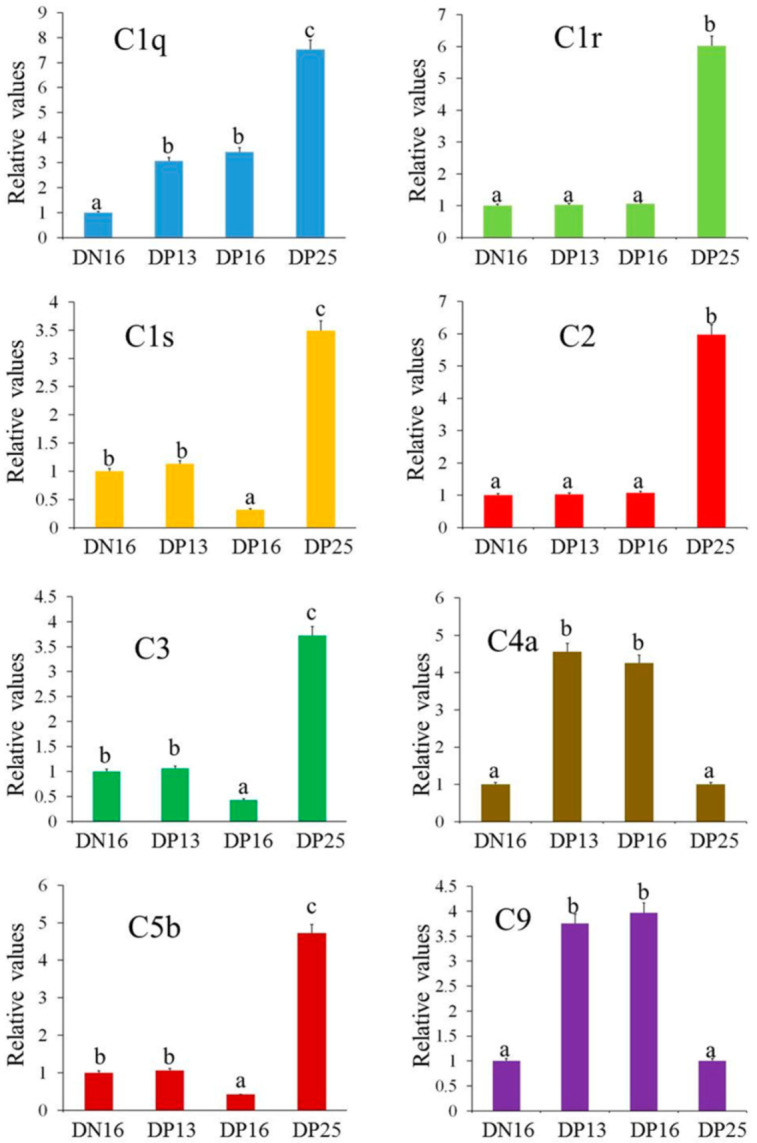
Relative expression levels of complement component mRNA in the spleen. Note: DN16 = day 16 of the estrous cycle; DP13 = day 13 of gestation; DP16 = day 16 of gestation; DP25 = day 25 of gestation. Significant differences (*p <* 0.05) are indicated by different letters.

**Figure 2 animals-11-03183-f002:**
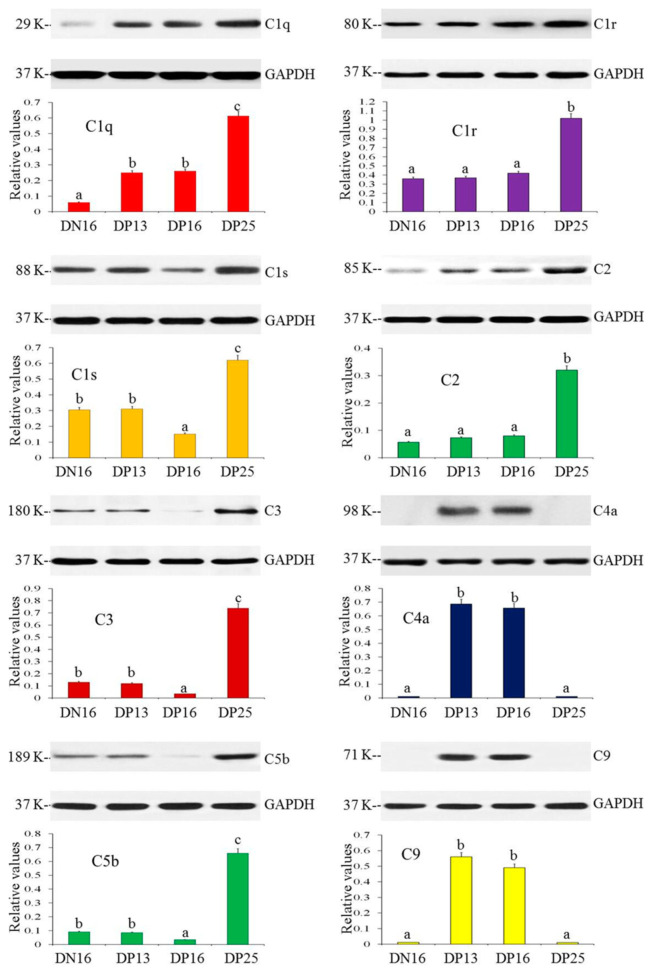
Protein expression of complement components in the spleen. Note: DN16 = day 16 of the estrous cycle; DP13 = day 13 of gestation; DP16 = day 16 of gestation; DP25 = day 25 of gestation. Significant differences (*p <* 0.05) are indicated by different letters.

**Figure 3 animals-11-03183-f003:**
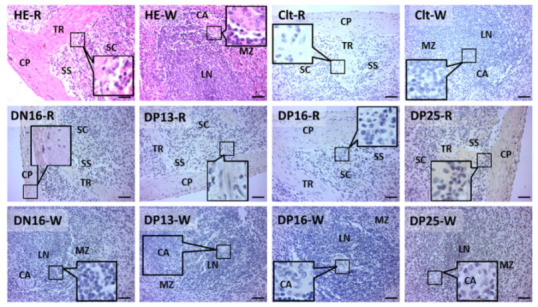
Immunohistochemical localization of C3 protein in the spleen. The spleen is divided into red pulp (R) and white pulp (W) and surrounds the thickened capsule (C). The capsule (CP) with several trabeculae (TR) projects into the substance of the spleen. Note: HE = stained by hematoxylin and eosin; Clt = negative control; SS = splenic sinuses; SC = splenic cords; MZ = marginal zone; LN = lymphoid nodule; CA = central arteriole; DN16 = day 16 of the estrous cycle; DP13 = day 13 of gestation; DP16 = day 16 of gestation; DP25 = day 25 of gestation. Bar = 50 µm.

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
