# Peer review of "Changes in Expression of Complement Components in the Ovine Spleen during Early Pregnancy"

_animals, 2021, doi:10.3390/ani11113183_

Round 1

Reviewer 1 Report

Regarding manuscript entitled “Changes in Expression of Complement Components in the Ovine Spleen during Early Pregnancy”, the authors aimed to detect the expression of complement components C1q, C1r, C1s, C2, C3, C4, C5 and C9 in maternal spleen during early pregnancy in sheep.

The study is interesting. However, major revision is important before its acceptance.

A clear aim must be added in the start of abstract section.

There are some important parts missed in the experimental design as following:

 “three groups (n = 6 for each group) were exposed to fertile rams and allowed to breed” this don’t mean that all these ewes are pregnant. How did the authors diagnose pregnancy and decide that this animals are pregnant or not?

“Days 13, 16, and 25 of pregnancy were chosen as the maternal spleens were under the effects of interferon-tau and/or P4 or not.” How?

Were all 18 ewes pregnant (100% in three groups)?

Were these animals synchronized for estrous? Please write the details. Without synchronization, samples weren’t collected on the same day and same environmental condition which may affect the evaluating parameters?

Were nutrition, age, BCS and parity of all these animals similar?

How did the authors obtain the spleen? Did they make biopsy, splenectomy, slaughter the ewe?

Was the expression of complement components C1q, C1r, C1s, C2, C3, C4, C5 and C9 in maternal spleen during early pregnancy in sheep affected by the number or sex of feti?

In figures “Significant differences (P < 0.05) are indicated by different letters.” Please use “a” for the highest or the lowest value (not as the authors did in C1s, C5b and…….).

The authors concluded that “Early pregnancy induced changes in expression of complement components C1q, C1r, C1s, C2, C3, C4, C5 and C9 in the maternal spleen.” Are these changes can be used as a signs of early pregnancy in ewes?

Author Response

Comments and Suggestions for Authors Regarding manuscript entitled “Changes in Expression of Complement Components in the Ovine Spleen during Early Pregnancy”, the authors aimed to detect the expression of complement components C1q, C1r, C1s, C2, C3, C4, C5 and C9 in maternal spleen during early pregnancy in sheep. The study is interesting. However, major revision is important before its acceptance. A clear aim must be added in the start of abstract section. Response: A clear aim had been added in the start of abstract section. There are some important parts missed in the experimental design as following: “three groups (n = 6 for each group) were exposed to fertile rams and allowed to breed” this don’t mean that all these ewes are pregnant. How did the authors diagnose pregnancy and decide that this animals are pregnant or not? Response: The experimental design had been revised. Please see the words in Animals and experimental design. “Days 13, 16, and 25 of pregnancy were chosen as the maternal spleens were under the effects of interferon-tau and/or P4 or not.” How? Response: The sentences had been revised. Please see the words in Animals and experimental design. We analysed the expression of complement components at these specific days of DN16, DP13, 16 and 25. The reasons are that the changed expression of the complement components is due to pregnancy, and the main factors are progesterone and IFNT during early pregnancy. The average of oestrous cycle is 17 days in sheep. There were significantly higher concentrations of progesterone on days 12-13 in plasma, and lower progesterone concentrations on 15-16 during the luteal phase of the ovine oestrous cycle (Mcnatty et al., 1973). IFNT (Protein X) and additional proteins were detected between days 14 and 21 in sheep (Godkin et al., 1982). There is no DN25, because the average of oestrous cycle is 17 days in sheep. DN13 is almost similar to DP13 according to above reasons. The above reasons had been present in previous reference that is cited in this manuscript. Godkin JD, Bazer FW, Moffatt J, et al. Purification and properties of a major, low molecular weight protein released by the trophoblast of sheep blastocysts at day 13-21. Journal of Reproduction & Fertility, 1982, 65(1):141. Mcnatty KP, Revefeim KJ, Young A. Peripheral plasma progesterone concentrations in sheep during the oestrous cycle. Journal of Endocrinology, 1973, 58(2):219-225. Were all 18 ewes pregnant (100% in three groups)? Response: The pregnancy rate after breeding is about 78%, and the ewes mated with intact rams without pregnancy were not used in this study, so we prepared 35 ewes in order to ensure six ewes on days 13, 16, and 25 of pregnancy, respectively. L. 109. What was pregnancy rate after breeding? Response: The pregnancy rate after breeding is about 78%. Were these animals synchronized for estrous? Please write the details. Without synchronization, samples weren’t collected on the same day and same environmental condition which may affect the evaluating parameters? Response: The information had been added. Please see the words in Animals and experimental design. Were nutrition, age, BCS and parity of all these animals similar? Response: The information had been added. Please see the words in Animals and experimental design. How did the authors obtain the spleen? Did they make biopsy, splenectomy, slaughter the ewe? Response: The spleen was obtained at slaughter. Was the expression of complement components C1q, C1r, C1s, C2, C3, C4, C5 and C9 in maternal spleen during early pregnancy in sheep affected by the number or sex of feti? Response: Sex differentiation in sheep fetuses has occurred on 35th day of pregnancy (Sabetghadam et al., 2018), so the expression of complement components in maternal spleen in this study did not affect by sex of fetus. In addition, and expression of complement components was not related to the number of fetus in this study, and in other study, the number of fetus is not considered for expression of interferon regulatory factor in the ovine Uterus (Fleming et al., 2009). Sabetghadam et al. Histo-morphologic study in fetuses and hormonal changes in fetal fluids during sex differentiation of sheep. Small Ruminant Research. 2018; 165:101-110 Fleming JA, Song G, Choi Y, Spencer TE, Bazer FW. Interferon regulatory factor 6 (IRF6) is expressed in the ovine uterus and functions as a transcriptional activator. Mol Cell Endocrinol. 2009; 299(2):252-260. In figures “Significant differences (P < 0.05) are indicated by different letters.” Please use “a” for the highest or the lowest value (not as the authors did in C1s, C5b and…….). Response: These had been revised. Please see the figures (C1s, C3 and C5b). The authors concluded that “Early pregnancy induced changes in expression of complement components C1q, C1r, C1s, C2, C3, C4, C5 and C9 in the maternal spleen.” Are these changes can be used as a signs of early pregnancy in ewes? Response: The statement had been added. Please see the words with yellow background in the end of the conclusion section.

Reviewer 2 Report

Introduction

This is well-written and sets the scene nicely for potential readers.

There are some sentences, which present basic knowledge and therefore are redundant, hence they should be deleted.

Most importantly however, the authors’ hypothesis and the objectives of the work are not clearly presented and they should be rewritten in better style.

Procedures

Table 1. Please move to supplementary material.

The details in 2.3. can be included in a table in supplementary material.

2.5. I am not sure that the results fitted with a normal distribution. Can the authors provide some evidence for that? Otherwise, they should redo the analysis with non-parametric tests (no big deal there!).

Results

Throughout the manuscript: please DO NOT give p values as a binary result. It decreases the value of the manuscript and it raises suspicions. In all cases, the precise p values should be provided (i.e., p = 0.0043). This must be changed throughout the text.

Figures 1 and 2. Please use colour to make better impressions to readers.

Figure 3. please use smaller size characters for the pictures. The current ones do not allow proper evaluation. This figure needs to be re-evaluated after improvement.

Discussion

The discussion will read better if divided in two subsections

Also, the authors need to add a paragraph about the clinical implications of this work for pre-conception management of ewes and the potential role of antibiotic / anthelminthic administration at the end of the lactation period.

The manuscript should be revised carefully and re-evaluated after submission of the corrected version.

Author Response

Comments and Suggestions for Authors

Introduction

This is well-written and sets the scene nicely for potential readers.

There are some sentences, which present basic knowledge and therefore are redundant, hence they should be deleted.

Response: Some sentences for basic knowledge had been deleted. Please see the words with yellow background in the Introduction section.

Most importantly however, the authors’ hypothesis and the objectives of the work are not clearly presented and they should be rewritten in better style.

Response: The hypothesis and the objectives had been revised. Please see the words with yellow background in the Introduction section.

Procedures

Table 1. Please move to supplementary material.

Response: The Table 1 had been moved to supplementary material as Table S1.

The details in 2.3. can be included in a table in supplementary material.

Response: The details for the antibodies had been included in Table S2 in supplementary material. Please see the supplementary material.

2.5. I am not sure that the results fitted with a normal distribution. Can the authors provide some evidence for that? Otherwise, they should redo the analysis with non-parametric tests (no big deal there!).

Response: The Statistical analysis section had been revised. Please see the words with yellow background in the Statistical analysis section.

Results

Throughout the manuscript: please DO NOT give p values as a binary result. It decreases the value of the manuscript and it raises suspicions. In all cases, the precise p values should be provided (i.e., p = 0.0043). This must be changed throughout the text.

Response: The p values had been provided. Please see the result section.

Figures 1 and 2. Please use colour to make better impressions to readers.

Response: The Figures 1 and 2 had been revised. Please see the Figures 1 and 2.

Figure 3. please use smaller size characters for the pictures. The current ones do not allow proper evaluation. This figure needs to be re-evaluated after improvement.

Response: The Figure 3 had been revised. Please see the Figure 3.

Discussion

The discussion will read better if divided in two subsections

Response: The discussion had been divided in two subsections. Please see the discussion section.

Also, the authors need to add a paragraph about the clinical implications of this work for pre-conception management of ewes and the potential role of antibiotic / anthelminthic administration at the end of the lactation period.

Response: A paragraph about the clinical implications of this work had been added. Please see the words with yellow background at the end of the discussion section.

The manuscript should be revised carefully and re-evaluated after submission of the corrected version.

Response: The manuscript had been revised carefully, and thank you for your suggestions.  

Reviewer 3 Report

The authors aimed to elucidate if the complement pathway participates in maternal splenic immune regulation during early pregnancy in sheep. To do so, they sampled ovine spleens on 16 day of the oestrous cycle, and days 13, 16, and 25 of gestation, and several techniques (qRT-PCR, western blot, and immunohistochemistry) were used to detect the changes in expression of complement components C1q, C1r, C1s, C2, C3, C4, C5 and C9 in the maternal spleens. They observed differential expression of complement components over time which suggests that complement pathways may be involved in modulating maternal splenic immune responses during early pregnancy in sheep.

Line

Text

Comment/suggestion to replace

36-37

The activity of the complement system is enhanced systemically, but complement inhibition is required for the pregnancy maintenance at the placenta during pregnancy [4].

Please review this sentence.

64-65

Days 13, 16, and 25 of pregnancy were chosen as the maternal spleens were under the effects of interferon-tau and/or P4 or not [14].

Please clarify.

88-97

with a mouse anti-C1q monoclonal antibody (Santa Cruz Biotechnology, Santa Cruz, CA, USA, sc-53544, 1:1000), a mouse anti-C1r monoclonal antibody (Santa Cruz Biotechnology, sc-514105, 1:1000), a mouse anti-C1s monoclonal antibody (Santa Cruz Biotechnology, sc-365273, 1:1000), a mouse anti-C2 monoclonal antibody (Santa Cruz Biotechnology, sc-373809, 1:1000), a mouse anti-C3 monoclonal antibody (Santa Cruz Biotechnology, sc-28294, 1:1000), a mouse anti-C4a monoclonal antibody (Santa Cruz Biotechnology, sc-271181, 1:1000), at 4 °C overnight.

with a mouse anti-C1q, anti-C1r, anti-C1s, anti-C2, anti-C3, anti-C4a, anti-C5b and anti-C9 monoclonal antibodies (Santa Cruz Biotechnology, Santa Cruz, CA, USA, sc-53544, sc-514105, sc-365273, sc-373809, sc-28294, sc-271181, sc-398247 and sc-390000, respectively; at 1:1000 each), at 4 °C overnight.

98

conjugatedanti-mouse secondary antibody

Conjugated anti-mouse secondary antibody

104

Table 1. Primer sequences for RT- qPCR analysis.

table 1 was been already published at:

-Zhang, L., Zhang, Q., Wang, H. et al. Effects of early pregnancy on the complement system in the ovine thymus. Vet Res Commun (2021). https://doi.org/10.1007/s11259-021-09837-9.

C1qA, C1r, C1s, C3, C4a C5b, and C9 genes accession numbers have new versions. If authors are willing to update the accession number, they should also verify if there are no changes in sizes.

104

Table 1. Primer sequences for RT- qPCR analysis.

Gene C2 – XM_027958809.1 is an obsolete version still available at NCBI, but the record has been removed as a result of standard genome annotation processing. Please see www.ncbi.nlm.nih.gov/genome/annotation_euk/process/ for more information.

105-112

Immunohistochemical analysis was only performed to detect C3 protein? Why?

Fig. 1 and 2

For C1s, C3, and C5b, please consider attributing a to the lowest values, b to intermediate values, and c to the highest values according to significant differences (P < 0.05)

159-160

I don’t understand the use of different colors for the columns neither in figure 1 nor in figure 2.

Significant differences (P < 0.05) are indicated by different letters within the same color column.

Significant differences (P < 0.05) are indicated by different letters.

163

Spleen is divided into red pulp and white pulp,

Spleen is divided into red pulp (R) and white pulp (W),

163-164

surrounds by thickened capsule (C). 163 Capsule (C) with several trabeculae (T) projects into the substance of the spleen.

Please clarify as in figure 3 appears CP, TR, and CA, but not C, T.

CP, TR, and CA have not been mentioned in the figure 3 legend.

169

C1q mRNA and protein upregulated

C1q mRNA, and protein were upregulated

Author Response

Comments and Suggestions for Authors

The authors aimed to elucidate if the complement pathway participates in maternal splenic immune regulation during early pregnancy in sheep. To do so, they sampled ovine spleens on 16 day of the oestrous cycle, and days 13, 16, and 25 of gestation, and several techniques (qRT-PCR, western blot, and immunohistochemistry) were used to detect the changes in expression of complement components C1q, C1r, C1s, C2, C3, C4, C5 and C9 in the maternal spleens. They observed differential expression of complement components over time which suggests that complement pathways may be involved in modulating maternal splenic immune responses during early pregnancy in sheep.

Lines 36-37, The activity of the complement system is enhanced systemically during pregnancy, but the activity is inhibited at the placenta for pregnancy maintenance [4]. Please review this sentence.

Response: These sentences had been revised. Please see the words with yellow background in Line 36.

Lines 64-65, Days 13, 16, and 25 of pregnancy were chosen as the maternal spleens were under the effects of interferon-tau and/or P4 or not [14]. Please clarify.

Response: These sentences had been revised. Please see the words with yellow background in Line 60.

Lines 88-97, with a mouse anti-C1q monoclonal antibody (Santa Cruz Biotechnology, Santa Cruz, CA, USA, sc-53544, 1:1000), a mouse anti-C1r monoclonal antibody (Santa Cruz Biotechnology, sc-514105, 1:1000), a mouse anti-C1s monoclonal antibody (Santa Cruz Biotechnology, sc-365273, 1:1000), a mouse anti-C2 monoclonal antibody (Santa Cruz Biotechnology, sc-373809, 1:1000), a mouse anti-C3. Please clarify. with a mouse anti-C1q, anti-C1r, anti-C1s, anti-C2, anti-C3, anti-C4a, anti-C5b and anti-C9 monoclonal antibodies (Santa Cruz Biotechnology, Santa Cruz, CA, USA, sc-53544, sc-514105, sc-365273, sc-373809, sc-28294, sc-271181, sc-398247 and sc-390000, respectively; at 1:1000 each), at 4 °C overnight.

Response: These antibodies had been included in Table S2. Please see the Table S2.

Line 98, conjugatedanti-mouse secondary antibody. Conjugated anti-mouse secondary antibody.

Response: These words had been revised. Please see the words with yellow background in Line 60.

Line 104, Table 1. Primer sequences for RT- qPCR analysis. table 1 was been already published at: Zhang, L., Zhang, Q., Wang, H. et al. Effects of early pregnancy on the complement system in the ovine thymus. Vet Res Commun (2021). https://doi.org/10.1007/s11259-021-09837-9.

C1qA, C1r, C1s, C3, C4a C5b, and C9 genes accession numbers have new versions. If authors are willing to update the accession number, they should also verify if there are no changes in sizes.

Response: The Table 1 had been moved to supplementary material as Table S1. The primer sequences were designed based on the primary  sequences of C1qA, C1r, C1s, C3, C4a C5b, and C9 genes accession numbers, and the results were based on the primer sequences. Therefore, it may be not necessary to update the accession number.

Line 104, Table 1. Primer sequences for RT- qPCR analysis. Gene C2 – XM_027958809.1 is an obsolete version still available at NCBI, but the record has been removed as a result of standard genome annotation processing. Please see www.ncbi.nlm.nih.gov/genome/annotation_euk/process/ for more information.

Response: The Gene C2 had been changed to XM_012100922.4 that is very similar to XM_027958809.1. Please see Table S1. 

Line 105-112, Immunohistochemical analysis was only performed to detect C3 protein? Why?

Response: C3 is a key protein for three complement pathways (classical, lectin and alternative pathways) (Janssen et al., 2005), so only C3 was investigated in the analysis of protein localization by immunohistochemistry. Other members of the ovine complement components have been investigated in mRNA and protein levels, which could explain the changing expression of complement pathways in ovine spleen during early pregnancy.

Janssen, B.J.; Huizinga, E.G.; Raaijmakers, H.C.; Roos, A.; Daha, M.R.; Nilsson-Ekdahl, K.; Nilsson, B.; Gros, P. Structures of complement component C3 provide insights into the function and evolution of immunity. Nature 2005, 437, 505-511.

Fig. 1 and 2, For C1s, C3, and C5b, please consider attributing a to the lowest values, b to intermediate values, and c to the highest values according to significant differences (P < 0.05).

Response: Fig. 1 and 2 had been revised. Please see Fig. 1 and 2.

Line 159-160, I don’t understand the use of different colors for the columns neither in figure 1 nor in figure 2.

Response: Fig. 2 had been revised. Please see the words with yellow background in Fig. 2.

Line 159-160, Significant differences (P < 0.05) are indicated by different letters within the same color column. Significant differences (P < 0.05) are indicated by different letters.

Response: These words had been revised. Please see the word with yellow background in Fig. 1 and 2.

Line 163, Spleen is divided into red pulp and white pulp. Spleen is divided into red pulp (R) and white pulp (W).

Response: These words had been revised. Please see the words with yellow background in Line 169.

Line 98, surrounds by thickened capsule (C). 163 Capsule (C) with several trabeculae (T) projects into the substance of the spleen. Please clarify as in figure 3 appears CP, TR, and CA, but not C, T. CP, TR, and CA have not been mentioned in the figure 3 legend.

Response: The Figure 3 had been revised. Please see the words with yellow background in the Figure 3.

Line 169, C1q mRNA and protein upregulated. C1q mRNA, and protein were upregulated.

Response: These words had been revised. Please see the words with yellow background in Line 177.

Round 2

Reviewer 1 Report

The authors improved their manuscript following the reviewers comments.

Reviewer 2 Report

The authors have taken into consideration the points raised and the suggestions provided. The manuscript has been improved and it reads better now.

Before final acceptance, moderate changes in English language are necessary, which be better for future readers. After that, the re-revised manuscript can be accepted.